# StreamBench: Towards Benchmarking Continuous Improvement of Language Agents

**Cheng-Kuang Wu**[1,2]*, **Zhi Rui Tam**[1]*, **Chieh-Yen Lin**[1], **Yun-Nung Chen**[1,2], **Hung-yi Lee**[2]

[1]Appier AI Research
[2]National Taiwan University
{brian.wu, ray.tam}@appier.com

## Abstract

Recent works have shown that large language model (LLM) agents are able to improve themselves from experience, which is an important ability for continuous enhancement post-deployment. However, existing benchmarks primarily evaluate their innate capabilities and do not assess their ability to improve over time. To address this gap, we introduce StreamBench, a pioneering benchmark designed to evaluate the continuous improvement of LLM agents over an input-feedback sequence. StreamBench simulates an online learning environment where LLMs receive a continuous flow of feedback stream and iteratively enhance their performance. In addition, we propose several simple yet effective baselines for improving LLMs on StreamBench, and provide a comprehensive analysis to identify critical components that contribute to successful streaming strategies. Our work serves as a stepping stone towards developing effective online learning strategies for LLMs, paving the way for more adaptive AI systems in streaming scenarios. Source code: `https://github.com/stream-bench/stream-bench`. Benchmark website: `https://stream-bench.github.io`.

## 1 Introduction

Recently, large-scale pretraining [1] and instruction fine-tuning [2] have driven paradigm shifts in how we interact with language models. These advancements allow us to use them out-of-the-box to solve problems. Consequently, many benchmarks have emerged to evaluate the general capabilities of these models. Some notable examples include MMLU [3], GSM8K [4], and BIG-Bench-Hard [5]. All these benchmarks aim to assess LLMs' *innate capabilities*, which we define as the general knowledge or reasoning abilities demonstrated when used out-of-the-box.

In addition to LLMs' strong *innate capabilities*, recent works have shown that LLM agents, which are LLMs augmented with extra components such as memory, retrievers, or tools, are able to improve themselves from experience. MemPrompt [6] shows that memory-enhanced GPT-3 can improve through time by storing past user feedback and retrieve them in the future. Reflexion [7] demonstrates that LLM agents can perform better in future trials by running repeated trials on the same dataset via self-reflection. ExpeL [8] further shows that LLM agents can learn from cross-task experience and improve performance without executing repeated trials on the target task.

Given LLM agents' self-improvement abilities, there remains a missing piece in the current evaluation landscape. Beyond measuring LLMs' *innate capabilities* with aforementioned *offline* benchmarks [3, 4, 5], it is important to assess their capacity to improve over time since we would like our systems to gradually improve after deployment. This gap motivated us to develop a new evaluation scenario–an *online* setting to measure LLM agents' ability to continuously enhance their performance over time.

---

*Equal contribution

38th Conference on Neural Information Processing Systems (NeurIPS 2024) Track on Datasets and Benchmarks.

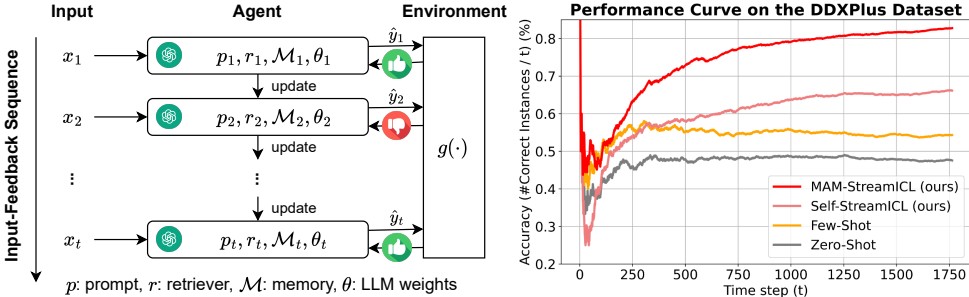

Figure 1: (Left) A schematic diagram showing the online evaluation setting of StreamBench, where agents update their components ($p, r, \mathcal{M}$, or $\theta$) from an input-feedback sequence to achieve the highest final accuracy (refer to Section 3.1 for details). (Right) Performance curve on the DDXPlus dataset on StreamBench. Agents are able to gradually improve with our proposed streaming baselines.

This *online* setting focuses on scenarios where LLM agents attempt to solve a specific downstream task and improve themselves from an input-feedback sequence, with the goal to maximize the accuracy for the whole sequence of the agent's predictions.

Given these rationales, we introduce StreamBench, a benchmark designed to evaluate LLM agents' ability to improve themselves over an input-feedback sequence. StreamBench simulates an environment where LLM agents are exposed to a sequence of users' natural language requirements and feedback. To the best of our knowledge, StreamBench is the first benchmark to evaluate LLM agents in streaming scenarios with a diverse range of tasks. StreamBench aims to inspire further efforts to develop more adaptive LLM agents, thereby enhancing their practical effectiveness. Our contributions can be summarized as follows:

- We introduce StreamBench, the first benchmark designed to evaluate LLM agents' ability to improve over an input-feedback sequence in an *online* setting across a wide range of tasks.

- We propose several simple yet effective baselines for enhancing LLM agents' performance in streaming scenarios, including a cost-effective multi-agent method that outperforms other baselines while maintaining the average cost of a single agent.

- We conduct analysis on the advantages and potential pitfalls of the proposed methods, providing insights into effective streaming strategies of LLMs.

## 2 Formulation

Consider a streaming scenario involving an agent, an external environment, and a sequence of inputs:

**Agent.** We define an *agent* as an LLM parameterized by $\theta$ and augmented with additional components to enhance the agent's capabilities, such as the external memory $\mathcal{M}$ and a retriever $r(\cdot)$ to store and retrieve useful information. Given an instance $x$ in natural language, a prompting template $p(\cdot)$, and a retrieval function $r(\cdot)$, the agent's output is denoted as $\hat{y} = f(p(x, r(\mathcal{M}))|\theta)$.

**Environment.** The external environment, denoted as $g(\cdot)$, provides feedback to the agent. The nature of $g(\cdot)$ varies depending on the specific downstream task and the type of feedback being collected. Potential roles for $g(\cdot)$ include human users, code execution environments, and API responses.

**Input-feedback sequence.** Consider a sequence of input stream where each input is denoted by $x_t$, with $t$ representing the $t$-th time step. After the agent provides the output $\hat{y}_t$, the environment provides feedback signal $fb_t = g(x_t, \hat{y}_t)$. Figure 1 shows an overview of the streaming scenario.

Algorithm 1 presents a simple framework for language agents to continuously learn from feedback. Benchmark users can adapt Algorithm 1 or develop their own algorithms to update components of their language agents, with the goal to maximize the accuracy of the entire sequence.

**Algorithm 1** Framework for Language Agents to Continuously Learn from Feedback on StreamBench

1: Initialize agent $f(\cdot|\theta)$, prompting template $p(\cdot)$, retriever $r(\cdot)$, and external memory $\mathcal{M}$;
2: **for** $t = 1, 2, \ldots, T$ **do**
3:     Receive instance $x_t$ from the data stream;
4:     The agent predicts $\hat{y}_t = f(p(x_t, r(\mathcal{M}))|\theta)$;
5:     Receive feedback signal $fb_t = g(x_t, \hat{y}_t)$;
6:     Update $p(\cdot)$, $r(\cdot)$, $\mathcal{M}$, or $\theta$ using $x_t$, $\hat{y}_t$, and $fb_t$; ▷ Benchmark users can develop their own algorithms for updating their Language Agents to learn continuously
7: **end for**

Traditionally, updating the agent at each time step $t$ involves updating the model parameters $\theta$. However, as foundation models grow increasingly larger, frequently updating the agent's network parameters has become computationally expensive. Recent advancements offer promising alternatives for iterative improvement by updating other components of the agent. For example, one can adapt existing iterative prompt refinement strategies to refine $p(\cdot)$ [9, 10, 11], update the weights of the retriever $r(\cdot)$ [12, 13, 14], expand the agent's memory $\mathcal{M}$ [6, 15], or use parameter-efficient fine-tuning techniques for augmenting $\theta$ [16]. These different strategies open new possibilities for continuous adaptation of agents without relying solely on network parameter updates. In this work, we develop several baselines for improving agents over time, with a particular focus on updating $p(\cdot)$ and $\mathcal{M}$. The baselines demonstrate both simplicity and effectiveness. We leave methods for updating $r(\cdot)$ and $\theta$, which require computationally expensive network parameter updates, for future research.

## 3 StreamBench

### 3.1 General setup

**Streaming sequence** Most public datasets are inherently static, meaning each instance does not have a time-related dependency. To adapt them for our streaming setup, we serialize each selected dataset in Section 3.2 by assigning a time step to each instance. To avoid arbitrary sequence assignment in the original datasets, we randomly shuffle each dataset using a fixed random seed. We release each dataset's assigned sequence obtained by this random seed in the supplementary materials to ensure reproducibility on StreamBench. Additionally, to ensure the robustness of our evaluation, we conduct experiments on different shuffled sequences with five random seeds, as discussed in Section 5.2. We also discuss the effects of distributional shifts in Appendix C.2.

**Feedback signals** Choosing appropriate type of feedback signal is a crucial consideration in StreamBench. Firstly, cost and practicality play a significant role; in practice, obtaining ground truth $y_t$ at each time step can be prohibitively expensive. For example, providing the exact code in programming tasks or the complete schema of each API call in tool use tasks is often impractical. In contrast, partial feedback, such as the helpfulness or correctness of the agent's output, is relatively easy to obtain–such as the "thumbs up" or "thumbs down" buttons commonly found in user interfaces of LLM applications. Given these rationales, we formalize the type of $fb_t$ as follows:

$$fb_t = g(x_t, \hat{y}_t), fb_t \in \{0, 1\}$$

where $fb_t$ is a scalar serving as a proxy for the correctness of $\hat{y}_t$ with respect to $x_t$, determined by the environment $g(\cdot)$ of the given downstream tasks. The feedback $fb_t \in \{0, 1\}$ is binary, indicating whether the agent's output $\hat{y}_t$ is correct. This simplified feedback setting aims to offer a unified evaluation framework for ensuring consistency and practicality across diverse tasks. We leave other designs of $fb_t$, such as ground truth or natural language feedback, for future works.

**Evaluation** In practice, an agent's goal is to satisfy as many user requirements as possible over a time sequence. We thus evaluate an agent by its aggregate metric at the final time step ($T$). For example, the final metric on a given dataset can be calculated as $\frac{\sum_{t=1}^{T} h(\hat{y}_t, y_t)}{T}$, where $h$ is the function for calculating the corresponding metric on a given dataset. Table 1 shows metrics for each dataset.

Table 1: Input, output, evaluation metrics, and number of testing instances of selected datasets.

| Task | Text-to-SQL | | | Python | Tool Use | Medical | QA |
|------|--------|-------|------|---------|----------|---------|-----|
| Dataset | Spider | CoSQL | BIRD | DS-1000 | ToolBench | DDXPlus | HotpotQA |
| Input ($x_t$) | Data requirements | | | Question | User query | Symptoms | Question |
| Output ($y_t$) | SQL code | | | Code | API calls | Diagnosis | Answer |
| Metric | Execution accuracy | | | Pass@1 | Accuracy | Accuracy | Exact Match |
| Test size ($T$) | 2,147 | 1,007 | 1,534 | 1,000 | 750 | 1,764 | 1,500 |

## 3.2 Datasets

To measure LLM agents' capacity for continuous improvement post-deployment, we select a diverse set of downstream tasks with potential real-world applications. Following the setting in Section 3.1, these tasks share the property that their ground truth output $y_t$ is costly to obtain at each time step.

**Text-to-SQL** For text-to-SQL tasks, the agent has to convert users' natural language queries into SQL code to meet their data requirements. StreamBench integrates three prominent datasets: Spider [17], CoSQL [18], and BIRD [19]. These datasets represent a progressive difficulty curve, allowing for evaluation of how well agents improve when faced with data of varying difficulties.

**Python programming** To evaluate coding ability improvement, we use the DS-1000 [20] dataset, which consists of real-world Python programming questions from StackOverflow. To successfully solve a given question, the agent must provide a solution and pass the associated test cases.

**Tool use** The ability to use external tools is a significant milestone in the development of LLMs, as it compensates for certain limitations, such as performing precise arithmetic operations or conducting web searches. For this purpose, we utilize the large-scale tool usage dataset ToolBench [21], and select the subset that includes stable and low-latency tool APIs collected in a previous work [22].

**Medical diagnosis** To assess LLMs' continuous improvement in applying expert knowledge, we use the DDXPlus [23] dataset, where agents must make a medical diagnosis out of 49 diagnoses based on patient profiles detailing their symptoms. This setup mimics how medical doctors improve their diagnostic skills through accumulated patient encounters. Evaluating LLMs on this dataset helps us understand their potential for continuous improvement in a highly specialized domain.

**Question answering** Question answering (QA) tasks evaluate an agent's ability to reason over supporting facts to answer users' questions. We adopt the distractor setting in HotpotQA [24], which requires reasoning over multiple supporting or distracting documents to answer questions. This helps measure the agent's improved proficiency in reasoning over grounded knowledge to provide accurate answers. Given the extensive volume of questions, we sampled 1,500 out of the total 7,410 questions.

Detailed information of the aforementioned datasets are provided in Table 1 and Appendix F.

## 4 Experiments

### 4.1 Baselines

A key objective of StreamBench is to compare the performance gains of LLM agents using *non-streaming* versus *streaming* methods. In *non-streaming* settings, methods focus on optimizing performance at a per-instance level, with improvements made independently for each testing instance. For these *non-streaming* methods, the overall performance boost on the testing set stems from improvements on individual testing instances. In contrast, *streaming* methods utilize information from past instances to improve future performance. For *streaming* methods, we adapt two previously proposed methods and introduce two new approaches to explore effective streaming strategies.

### 4.1.1 Non-streaming methods

**Zero-shot**   It reflects the basic instruction-following abilities of LLMs for solving a given task.

**Few-shot**   It involves providing several ground truth $(x, y)$ pairs in the prompting template $p(\cdot)$. For datasets with training sets, we construct few-shot examples from the training data. For datasets without training sets, we compose few-shot examples and inspect their quality to ensure reliability. We include few-shot examples for each dataset in the supplementary materials for reproducibility.

**Chain-of-thought (CoT)**   Following previous work [25], we employ a trigger phrase (e.g., "Let's think step by step.") to instruct the LLM to generate the reasoning process before providing its final answer. We then extract the answer in the correct format from the generated reasoning text.

**Self-Refine**   Self-Refine [26] is a technique where the LLM iteratively improves its output based on self-feedback. The model generates an initial response and refines it through multiple iterations of refinement. It leverages LLMs' ability to self-evaluate and adjust its responses at a per-instance level.

### 4.1.2 Streaming methods

**GrowPrompt**   We adapt the previously proposed method GrowPrompt [6], where $(x_t, \hat{y}_t, fb_t)$ of the latest time steps are stored in a sliding window $\mathcal{W}$. The contents of $\mathcal{W}$ are incorporated into the prompt at inference time to output $y_t = f(p(x_t, \mathcal{W})|\theta)$. This provides the agent with information from the past $k$ instances, where $k$ is a hyperparameter. Since LLM agents take text input, we verbalize $fb_t \in \{0, 1\}$ to inform the agent of whether its output $y_t$ correctly satisfies the input $x_t$.

**MemPrompt**   As an advanced version of GrowPrompt, MemPrompt [6] incorporates an external memory $\mathcal{M}$ to store $(x_t, \hat{y}_t, fb_t)$ of all past time points. During inference, a retriever $r(\cdot)$ is used to select $k$ elements from $\mathcal{M}$, and $fb_t$ is also verbalized to inform the agent of $\hat{y}_t$'s correctness.

**Self-StreamICL**   Previous works [27, 28] have found that incorrect examples can negatively impact in-context learning (ICL) performance, though the extent varies for different LLMs. Based on these insights, we hypothesize that while GrowPrompt and MemPrompt use verbalized $fb_t$ to inform the agent about the correctness of its output, incorrect $\hat{y}_t$ still introduces distracting signals that can hinder improvement. Therefore, we propose to save $(x_t, \hat{y}_t)$ pairs to memory $\mathcal{M}$ *only when $fb_t = 1$, eliminating the need to save verbalized $fb_t$.* This method, called Self-StreamICL, operates similarly to regular ICL, except that the labels are now self-generated and gradually accumulate over the data stream, without the need to preconstruct few-shot examples. For more details, refer to Algorithm 2.

**Multi-Agentic-Memory StreamICL**   In Self-StreamICL, the agent learns exclusively from its own past experiences. However, we hypothesize that different LLM agents possess distinct strengths and weaknesses, so they can potentially benefit from the experiences of other agents. To explore this idea, we introduce Multi-Agentic-Memory StreamICL (MAM-StreamICL), which employs a multi-agent framework where multiple LLM agents share a common memory. This shared memory incorporates the past outputs of all agents, allowing each agent to learn from the diverse experiences of the others.

We implement a simple round-robin-like scheduling to switch between different LLM agents outlined in Algorithm 2. This ensures that each agent contributes to the shared memory in a balanced manner. Our experiments show that this straightforward strategy can boost performance beyond the average performance of the individual agents. In fact, Self-StreamICL can be seen as a special case of MAM-StreamICL with only one LLM agent.

Note that the high cost associated with scaling is the most critical drawback of multi-agent methods proposed by previous works, and the key advantage of MAM-StreamICL is its cost-effectiveness. Unlike methods such as Multiagent Debate [29] and RECONCILE [30], the cost of MAM-StreamICL does not scale proportionally with the number of agents. Instead, *the cost is equivalent to the averaged cost of a single agent, since only one agent is assigned to answer at each time step.*

**Algorithm 2** Round-Robin Algorithm for MAM-StreamICL
___
1: Initialize $K$ agents $f_0(\cdot|\theta_0), f_1(\cdot|\theta_1), ..., f_{K-1}(\cdot|\theta_{K-1})$; ▷ K = 1 in the Self-StreamICL baseline
2: Initialize prompt $p(\cdot)$, retriever $r(\cdot)$, and external memory $\mathcal{M}_0$, all shared between agents;
3: **for** $t = 1, 2, \ldots, T$ **do**
4:     Receive instance $x_t$ from the data stream;
5:     Select the next agent by $k = t \bmod K$;
6:     The $k$-th agent predicts $\hat{y}_t = f_k(p(x_t, r(\mathcal{M}_{t-1}))|\theta_k)$;
7:     Receive feedback signal $fb_t = g(x_t, \hat{y}_t)$;     ▷ $fb_t \in \{0, 1\}$ under the StreamBench setting
8:     **if** $fb_t = 1$ **then**                     ▷ which means the self-output $\hat{y}_t$ is correct
9:         $\mathcal{M}_t \leftarrow \mathcal{M}_{t-1} \cup \{(x_t, \hat{y}_t)\}$;
10:     **else**
11:         $\mathcal{M}_t \leftarrow \mathcal{M}_{t-1}$;
12:     **end if**
13: **end for**
___

## 4.2 Implementation details

We conduct experiments using three LLM families: GPT [31, 32], Gemini [33, 34], and Claude [35]. For our main experiments, we use the endpoints `gpt-3.5-turbo-0125`, `gemini-1.0-pro-001`, and `claude-3-haiku-20240307`. These models represent cost-effective LLMs, balancing performance and affordability. The models initialize the $K = 3$ agents in MAM-StreamICL. For methods with $\mathcal{M}$ (MemPrompt, Self-StreamICL, and MAM-StreamICL), we implement $\mathcal{M}$ as a vector database. We use `BAAI/bge-base-en-v1.5` to encode $x_t$ as key embeddings and save $x_t, \hat{y}_t$ (and $fb_t$ for MemPrompt) as values. For hyperparameters and prompts, refer to Appendix A and F.

## 4.3 Main results

The main results are shown in Table 2, which lists the averaged performance of the three LLM agents. The only exception is MAM-StreamICL, which only runs once on the streaming sequence where each agent takes turns at each time step. For results of each respective model, refer to Appendix B.

Overall, streaming methods outperform non-streaming methods, though the extent of improvement varies across different datasets. These results demonstrate the value of leveraging input-feedback streams to enhance agent performance on downstream tasks. In addition, as demonstrated by the robust performance of Self-StreamICL compared to GrowPrompt and MemPrompt, leveraging feedback as simple as correctness can enable agents to improve even more through self-generated outputs $\hat{y}_t$. This provides an important insight: rather than solely focusing on prompting pipelines to boost per-instance performance, adopting simple yet effective streaming approaches, such as collecting correctness feedback, could potentially lead to notable improvements on LLM agents. Lastly, MAM-StreamICL shows the most notable and consistent performance boost across all datasets.

Table 2: Averaged performance of three LLM agents across different baselines and datasets.

| Task | Text-to-SQL | | | Python | Tool Use | Medical | QA |
|---|---|---|---|---|---|---|---|
| **Dataset** | **Spider** | **CoSQL** | **BIRD** | **DS-1000** | **ToolBench** | **DDXPlus** | **HotpotQA** |
| *Non-streaming* | | | | | | | |
| Zero-Shot | 67.89 | 50.55 | 29.60 | 37.70 | 61.38 | 52.85 | 48.49 |
| Few-Shot | 68.55 | 50.61 | 30.40 | 33.33 | 68.58 | 60.98 | 53.11 |
| CoT | 61.53 | 46.01 | 27.23 | 25.93 | 58.98 | 58.20 | 52.47 |
| Self-Refine | 67.75 | 49.49 | 29.62 | 36.30 | 60.67 | 52.89 | 43.53 |
| *Streaming* | | | | | | | |
| GrowPrompt | 69.90 | 51.97 | 30.35 | 33.77 | 65.07 | 55.10 | 51.38 |
| MemPrompt | 70.78 | 53.29 | 31.99 | 35.47 | 64.31 | 54.02 | 52.62 |
| Self-StreamICL | 74.63 | 55.05 | 35.31 | 41.30 | 71.33 | 70.56 | 54.80 |
| MAM-StreamICL | **75.69** | **55.17** | **36.38** | **43.10** | **75.87** | **83.50** | **55.20** |

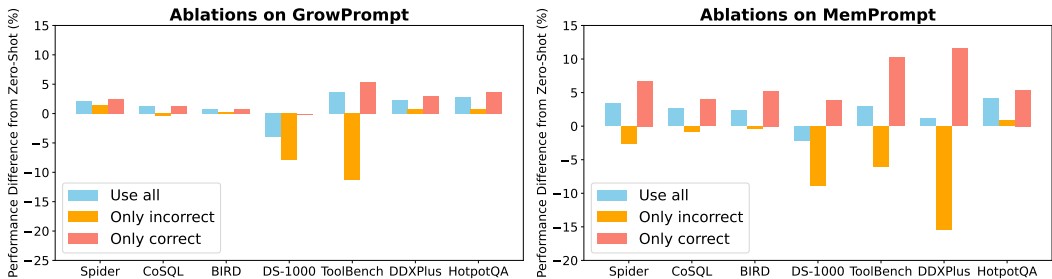

Figure 2: Correctness ablations. The y-axis denotes performance difference from zero-shot. The results are the average of three LLM endpoints. Please refer to Appendix D for results of each LLM.

## 5 Discussion

### 5.1 What makes effective streaming strategies?

This subsection provides insights into the key aspects that contribute to successful streaming strategies. We identify two effective factors for streaming improvements as follows:

#### 5.1.1 Collecting correct self-output

To investigate whether incorrect self-output hinders agents' improvement, we conducted ablation studies on GrowPrompt and MemPrompt. In the default setting in Table 2, both methods use all $k$ retrieved $(x_t, \hat{y}_t, fb_t)$ triples during inference (use all). In contrast, the ablations either use only the triples where $fb_t = 0$ (only incorrect), or use only the triples where $fb_t = 1$ (only correct).

The ablation results in Figure 2 reveal several findings. First, using incorrect self-output degrades performance, sometimes even worse than the zero-shot baseline. In contrast, using only correct self-output consistently boosts performance over the zero-shot baseline, with particularly consistent improvements observed in the MemPrompt (only correct) method. An important observation is that, even if $fb_t$ is verbalized to inform the agent whether its $\hat{y}_t$ correctly satisfies $x_t$ in GrowPrompt and MemPrompt, simply informing the agent that its self-output is incorrect does not help it learn from past mistakes. Conversely, telling the LLM agent what it does correctly is very effective in facilitating improvement. These findings underscore the importance of collecting and utilizing correct self-output in streaming. This also explains the intuition and effectiveness behind Self-StreamICL, where input-output pairs are saved to $\mathcal{M}$ only when the self-output is correct.

#### 5.1.2 Sharing memory across multiple agents

Another key insight is the benefit of sharing memory across multiple agents, as demonstrated in MAM-StreamICL. To analyze why memory sharing works, we use DDXPlus as an example and visualize the confusion matrices for a subset of diagnoses related to upper respiratory tract diseases.

Figure 3 presents the confusion matrices for three different LLM agents: `gpt-3.5-turbo-0125`, `gemini-1.0-pro`, and `claude-3-haiku-20240307`, along with the matrix of MAM-StreamICL. Each matrix illustrates the proficiency of an agent across various medical diagnosis categories. It is evident that each model excels in certain areas while struggling in others. For instance, `gpt-3.5-turbo-0125` shows high accuracy in predicting "acute rhinosinusitis" and "allergic sinusitis" but struggles with "chronic rhinosinusitis" and "URTI". In contrast, `gemini-1.0-pro` performs well in "URTI", and `claude-3-haiku` could solve "chronic rhinosinusitis".

The diversity in performance across models suggests that their collective past experiences can provide complementary strengths, thereby enhancing overall performance when these experiences are shared. Since each agent takes turn to solve an incoming $x_t$ at each time point $t$, the shared memory system allows the agents to benefit from others while maintaining a cost similar to that of a single agent. We also conduct further ablation studies in Appendix D to discuss the importance of sharing memory.

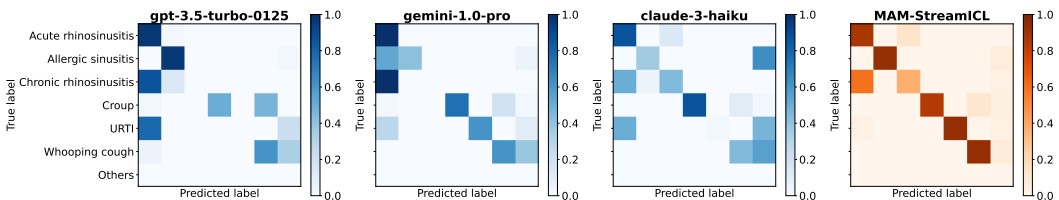

Figure 3: Confusion matrices of the diagnoses subset of upper respiratory tract diseases in DDXPlus.

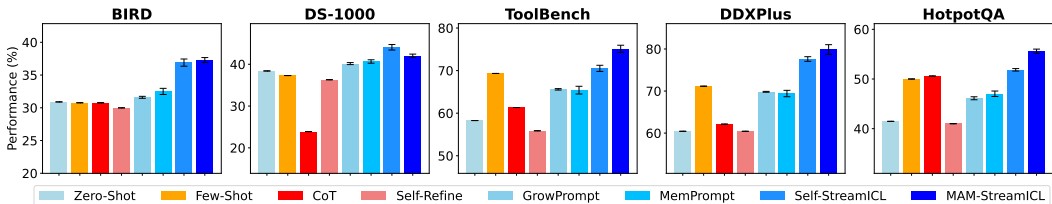

Figure 4: Averaged performance and standard errors of each method on five shuffled sequences.

## 5.2 Robustness to different streaming sequences

Given the time-variant nature of streaming, evaluating each method's robustness across different data streams is essential. Therefore, we rerun the streaming baselines with 5 random seeds on five tasks. Figure 4 presents the averaged performance and standard errors of `claude-3-haiku` across 5 shuffled sequences, with results for `gpt-3.5-turbo` and `gemini-1.0-pro` provided in Appendix C. The performance ranking of streaming baselines remains mostly consistent across datasets, with Self-StreamICL and MAM-StreamICL being the top performers. Due to the high cost of running all 5 sequences on StreamBench, we select a fixed sequence for fair comparison among future benchmark users. However, we also release all 5 sequences for those who wish to conduct a thorough evaluation.

## 5.3 Would stronger LLMs still benefit from streaming?

To evaluate whether stronger LLMs still benefit from streaming, we tested two newer models: `gpt-4o-2024-08-06` and `gemini-1.5-flash-001`. Due to the high cost, we only run the methods shown in Table 3. We found that with Self-StreamICL, these stronger models still showed significant performance improvements. This demonstrates that even the most advanced models can leverage the information from streaming data to further enhance their performance across diverse tasks.

## 5.4 Cost analysis

For benchmark users to estimate the cost, the token usage of all baselines is listed in Appendix E.

Table 3: Performance of `gpt-4o-2024-08-06` and `gemini-1.5-flash-001` on StreamBench.

| Task | Text-to-SQL | | | Python | Tool Use | Medical | QA |
|---|---|---|---|---|---|---|---|
| Dataset | Spider | CoSQL | BIRD | DS-1000 | ToolBench | DDXPlus | HotpotQA |
| *gemini-1.5-flash* | | | | | | | |
| Zero-shot | 69.63 | 48.26 | 33.83 | 50.20 | 69.47 | 58.90 | 60.60 |
| Few-shot | 71.40 | 49.35 | 37.03 | 50.60 | 72.13 | 73.58 | 64.87 |
| CoT | 72.52 | 52.73 | 35.14 | 44.80 | 68.00 | 64.06 | 63.13 |
| Self-StreamICL | **77.83** | **56.21** | **41.20** | **52.20** | **75.07** | **86.34** | **65.20** |
| *gpt-4o-2024-08-06* | | | | | | | |
| Zero-shot | 73.54 | 53.33 | 34.42 | 54.90 | 72.40 | 70.64 | 65.53 |
| Few-shot | 76.85 | 57.60 | 36.25 | 52.30 | 71.47 | 83.45 | 66.87 |
| CoT | 72.52 | 54.82 | 31.16 | 41.90 | 66.80 | 73.02 | 62.80 |
| Self-StreamICL | **80.58** | **59.19** | **42.63** | **59.40** | **76.27** | **92.01** | **67.00** |

# 6 Related work

## 6.1 Online learning

Online learning [36] explores the incremental updating of models as new data arrives, making it valuable for dynamic improvement in downstream applications. Traditionally, it focuses updating network weights, such as in methods for training recurrent neural networks [37], online representation learning for image classification [38], and adapting language models to learn new world knowledge [39]. Recent advancements have introduced strategies for improving LLM agents by updating prompts [9, 10, 11], memory [6, 15], or retrievers [12, 13, 14]. These new strategies are promising for designing new algorithms to adapt LLMs in the online setting. However, there are no standard testbeds for this setup. Addressing this gap, we propose StreamBench, the first benchmark to pave the way for developing more dynamic adaptation methods for LLM agents.

## 6.2 Improvement from feedback with LLMs

Recent works have shown that LLMs can improve from feedback when augmented with prompting pipelines or memory mechanisms, forming two main research branches. One is instance-level improvement methods, such as ReAct [40], Self-ICL [41], and Self-Refine [26]. These methods focus on boosting performance on each input instance without leveraging information from past instances. The other is time-sequence-level improvement methods. For example, MemPrompt [6] enhances GPT-3 by storing past user feedback and retrieve them in the future. Reflexion [7] shows that LLM agents can perform better in future trials by running repeated trials on the same dataset, but this is not always practical in real-world scenarios. ExpeL [8] shows that LLM agents can benefit from cross-task experience without needing repeated trials on the target task. However, these works use different datasets and lack a standardized evaluation setting. StreamBench bridges this gap by providing a consistent empirical testbed across diverse tasks to evaluate LLM agents' improvement.

# 7 Conclusion

In this work, we introduce a new evaluation setting to measure LLM agents' performance improvement on downstream tasks, and propose StreamBench as an instance of this setting. There are two major findings in our experiments. Firstly, collecting correct self-generated outputs improve performance consistently, while informing agents of their incorrect outputs sometimes degrade performance. Secondly, sharing memory across multiple agents is a promising cost-effective technique, as MAM-StreamICL achieves robust performance while maintaining the average cost of a single agent.

StreamBench serves as a stepping stone towards more adaptive AI systems. Future directions include exploring *online active learning* where agents could inquire feedback only when necessary, or viewing multi-agent collaboration as multi-arm bandits (MABs) to develop more sophisticated methods for selecting agents and sharing memory at each time point. It is also practical to investigate the utilization of different feedback signals beyond correctness, such as users' natural language feedback. We hope that this work inspires development of adaptive methodology for improving LLM agents.

# 8 Limitations

**Tasks and modality coverage** The current version of StreamBench includes tasks such as programming, text-to-SQL conversion, medical diagnosis, question-answering, and tool use. While diverse, they do not encompass all possible types of tasks or domains where LLMs can be applied. StreamBench is also limited to text and does not cover other modalities such as image and audio.

**Sim2Real gap** Although we have attempted to simulate feedback signals as practical as possible, there may still be a gap between the simulated correctness feedback in StreamBench and the feedback encountered in real-world applications. Real-world feedback can be more diverse, noisy, and context-dependent, which may not be fully captured by the current benchmark.

## Acknowledgments and Disclosure of Funding

We would like to express our gratitude to Chih-Han Yu, Wei-Lin Chen, Yi-Lin Tsai, Chao-Chung Wu, Zhen-Ting Liu, and An-Zi Yen for their valuable comments and feedback on this work. Their insights greatly contributed to the improvement of our research. We declare no competing interests related to this work.

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

# A Hyperparameters

For decoding strategies of all model endpoints used in this work, we set *temperature* to $0$ and *top-p* to $1$. The few-shot baseline and streaming baselines (GrowPrompt, MemPrompt, Self-StreamICL, and MAM-StreamICL) incorporate information from $k$ instances into the prompt $p(\cdot)$ to improve LLM agents. We use the same $k$ across these baselines for fair comparison. We set $k = 16$ for Spider, CoSQL, BIRD, ToolBench, and DDXPlus. For DS-1000 and HotpotQA, we set $k = 4$ to avoid exceeding the context size of `gpt-3.5-turbo-0125`.

We also analyze how different text embeddings used in memory correlate with streaming performance in Table 4. We observe that within the same text encoder family (bge), the larger model (109M parameters) generally delivers better performance. However, smaller models (22.7M parameters) can also achieve strong results, indicating that each LLM may benefit from a specific encoder.

Table 4: Performance of Self-StreamICL (implemented with different text encoders and LLMs) on the DDXPlus dataset.

| Text encoder / LLMs | gpt-3.5-turbo-0125 | claude-3-haiku | gemini-1.5-flash-001 |
|---|---|---|---|
| all-MiniLM-L6-v2 (22.7M) | 63.61 | **78.91** | 83.50 |
| bge-small-en-v1.5 (33.4M) | 63.55 | 75.51 | 83.90 |
| bge-base-en-v1.5 (109M) | **66.16** | 76.02 | **86.34** |

# B Main results of each LLM endpoint

Main experiments results for three different LLM families models are shown below:

Table 5: Performance of different baselines and datasets for `gpt-3.5-turbo-0125`

| Task | Text-to-SQL | | | Python | Tool Use | Medical | QA |
|---|---|---|---|---|---|---|---|
| Dataset | Spider | CoSQL | BIRD | DS-1000 | ToolBench | DDXPlus | HotpotQA |
| *Non-streaming* | | | | | | | |
| Zero-Shot | 68.89 | 52.83 | 29.75 | 41.50 | 64.13 | 47.56 | 54.53 |
| Few-Shot | 69.54 | 52.73 | 28.94 | 33.30 | 70.13 | 54.31 | 54.93 |
| CoT | 65.53 | 47.96 | 29.21 | 32.80 | 57.20 | 53.18 | 57.13 |
| Self-Refine | 67.21 | 51.64 | 29.92 | 39.80 | 64.53 | 47.68 | 40.06 |
| *Streaming* | | | | | | | |
| GrowPrompt | 70.89 | 53.43 | 30.31 | 27.20 | 67.60 | 44.62 | 55.80 |
| MemPrompt | 73.68 | 54.32 | 34.16 | 29.40 | 67.07 | 51.53 | 56.73 |
| StreamICL | 75.59 | 54.92 | 35.07 | 41.90 | 74.00 | 66.16 | 55.80 |

Table 6: Performance of different baselines and datasets for `gemini-1.0-pro-001`

| Task | Text-to-SQL | | | Python | Tool Use | Medical | QA |
|---|---|---|---|---|---|---|---|
| Dataset | Spider | CoSQL | BIRD | DS-1000 | ToolBench | DDXPlus | HotpotQA |
| *Non-streaming* | | | | | | | |
| Zero-Shot | 68.28 | 49.26 | 28.16 | 33.20 | 61.73 | 50.57 | 49.47 |
| Few-Shot | 68.33 | 49.65 | 31.49 | 29.40 | 66.27 | 57.48 | 54.40 |
| CoT | 52.31 | 40.81 | 21.71 | 21.10 | 58.40 | 59.30 | 49.67 |
| Self-Refine | 69.59 | 46.47 | 28.94 | 32.80 | 61.60 | 50.57 | 49.53 |
| *Streaming* | | | | | | | |
| GrowPrompt | 71.59 | 52.43 | 28.68 | 33.10 | 61.87 | 51.13 | 52.80 |
| MemPrompt | 70.28 | 54.22 | 30.18 | 35.50 | 58.80 | 43.59 | 55.00 |
| StreamICL | 76.48 | 55.41 | 33.25 | 35.80 | 68.80 | 69.50 | 57.20 |

Table 7: Performance of different baselines and datasets for `claude-3-haiku-20240307`

| Task | Text-to-SQL | | | Python | Tool Use | Medical | QA |
|---|---|---|---|---|---|---|---|
| Dataset | Spider | CoSQL | BIRD | DS-1000 | ToolBench | DDXPlus | HotpotQA |
| *Non-streaming* | | | | | | | |
| Zero-Shot | 66.51 | 49.55 | 30.90 | 38.40 | 58.27 | 60.43 | 41.47 |
| Few-Shot | 67.77 | 49.45 | 30.77 | 37.30 | 69.33 | 71.15 | 50.00 |
| CoT | 66.74 | 49.26 | 30.77 | 23.90 | 61.33 | 62.13 | 50.60 |
| Self-Refine | 66.46 | 50.35 | 29.99 | 36.30 | 55.87 | 60.43 | 41.00 |
| *Streaming* | | | | | | | |
| GrowPrompt | 67.21 | 50.05 | 32.07 | 41.00 | 65.73 | 69.56 | 45.53 |
| MemPrompt | 68.37 | 51.34 | 31.62 | 41.50 | 67.07 | 66.95 | 46.13 |
| Self-StreamICL | 71.82 | 54.82 | 37.61 | 46.20 | 71.20 | 76.02 | 51.40 |

# C   Robustness to different streaming sequences

## C.1   Performance on sequences shuffled by five different random seeds

Results of averaged performance and standard errors of `gpt-3.5-turbo-0125`, `gemini-1.0-pro-001`, and `claude-3-haiku-20240307` are listed below:

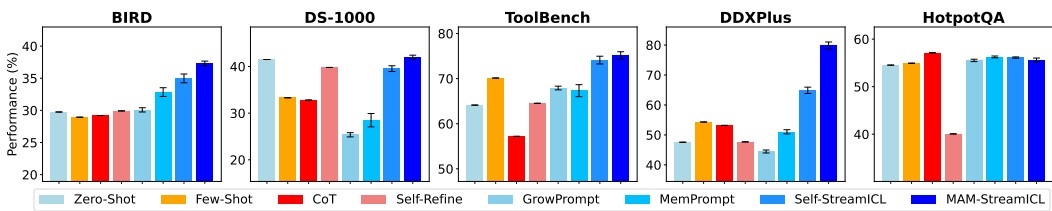

Figure 5: Averaged performance and standard errors of `gpt-3.5-turbo` on five shuffled sequences.

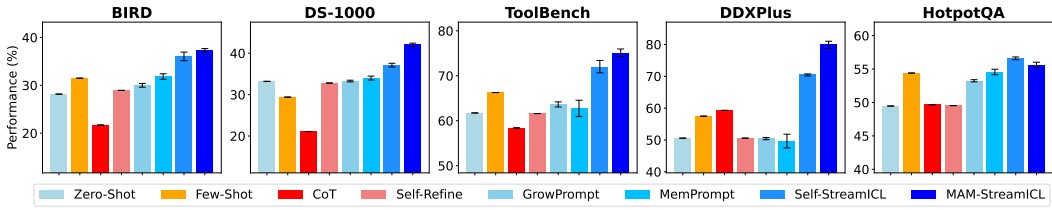

Figure 6: Averaged performance and standard errors of `gemini-1.0-pro` on five shuffled sequences.

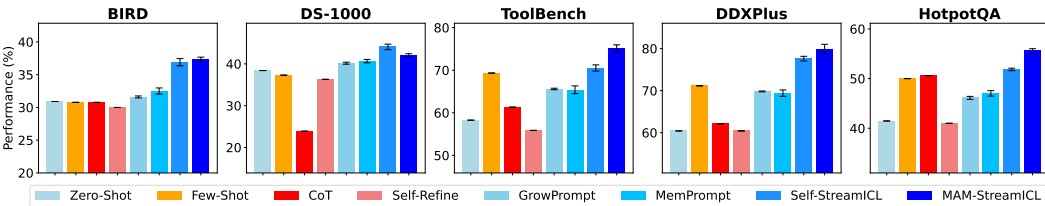

Figure 7: Averaged performance and standard errors of `claude-3-haiku` on five shuffled sequences.

## C.2 Performance on the sequence with distributional shifts

Since we randomly assign time steps to each instance, our main results in Table 2 simulates scenarios where each instance in the streaming sequence is drawn from the same distribution. However, it is important to investigate how well streaming methods perform on sequences with distributional shifts. To this end, we conduct experiments on BIRD [19] by arranging instances from the same database consecutively (DB1, DB2, ..., DB11), resulting in 10 distributional shifts during streaming. The results are shown in Table 8. The key finding is that Self-StreamICL still outperforms non-streaming baselines, and the method's performance does not differ drastically when distributional shifts occur.

Table 8: Performance of Self-StreamICL with different LLMs on BIRD.

| Method / LLM | gpt-3.5 | gemini-1.0-pro | claude-3-haiku | gemini-1.5-flash | gpt-4o |
|---|---|---|---|---|---|
| Zero-Shot | 29.75 | 28.16 | 30.90 | 33.83 | 34.42 |
| Few-Shot | 28.94 | 31.49 | 30.77 | 37.03 | 36.25 |
| CoT | 29.21 | 21.71 | 30.77 | 35.14 | 31.16 |
| Self-StreamICL (no shifts) | 35.07 | 33.25 | 37.61 | 41.20 | 42.63 |
| Self-StreamICL (w/ shifts) | 36.31 | 32.60 | 36.57 | 40.48 | 43.09 |

## D  Detailed ablation study results

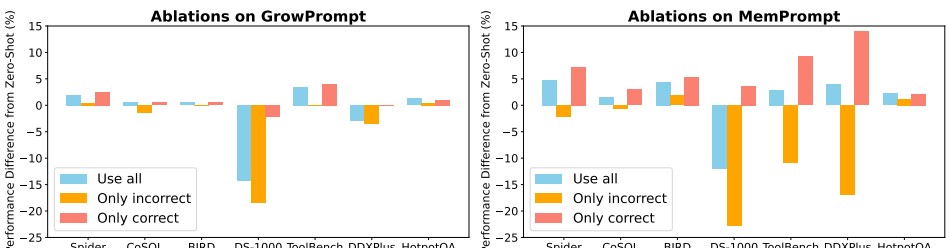

Figure 8: Correctness ablations of `gpt-3.5-turbo-0125`.

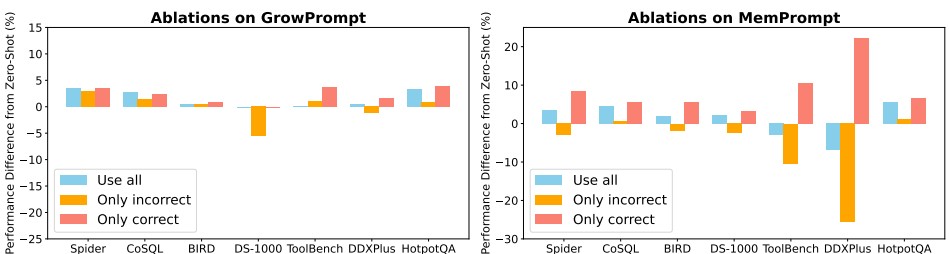

Figure 9: Correctness ablations of `gemini-1.0-pro-001`.

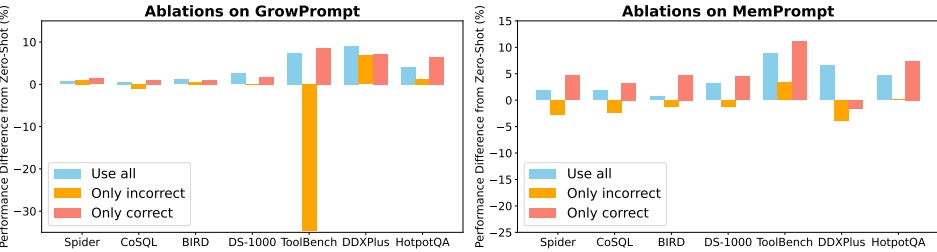

Figure 10: Correctness ablations of `claude-3-haiku-20240307`.

Table 9: Ablation studies with MAM-StreamICL on DDXPlus. The ablated version of MAM-StreamICL only uses memory of the single corresponding agent, while still uses round-robin algorithm for multi-agent inference. We can see that both multi-agent memory and inference are beneficial for performance boost. The detailed algorithm for this ablation study can be found in Appendix F.

| Method | GPT | Gemini | Claude | Memory | Inference |
|--------|-----|--------|--------|--------|-----------|
| Zero-Shot | 47.56 | 50.57 | 60.43 | x | single agent |
| Self-StreamICL | 66.16 | 69.50 | 76.02 | single agent | single agent |
| MAM-StreamICL (ablation) | 65.31 | 72.05 | 81.52 | single agent | multi agent |
| MAM-StreamICL | **83.50** | **83.50** | **83.50** | multi agent | multi agent |

# E   Token cost breakdown for each LLM endpoint

Table 10, 11, 12, and 13 shows how many millions of input, output tokens are used by `gpt-3.5-turbo-0125`, `gemini-1.0-pro-001`, `claude-3-haiku-20240307`, and the latest models. The cost of MAM-StreamICL is simply the averaged cost of the three LLMs due to the round-robin algorithm. As most LLM endpoints use input and output tokens to charge usage fees, we provide this information for benchmark users to estimate the cost for running StreamBench.

Table 10: Cost analysis (using millions of input and output tokens of `gpt-3.5-turbo-0125`).

| Task | Text-to-SQL | | | Python | Tool Use | Medical | QA |
|------|-------|-------|------|--------|----------|---------|-----|
| Dataset | Spider | CoSQL | BIRD | DS-1000 | ToolBench | DDXPlus | HotpotQA |
| *Non-streaming* | | | | | | | |
| Zero-Shot | 0.734/0.064 | 0.392/0.023 | 1.338/0.070 | 0.522/0.053 | 5.202/0.019 | 1.131/0.013 | 2.182/0.014 |
| Few-Shot | 2.367/0.061 | 0.954/0.021 | 3.206/0.120 | 1.984/0.074 | 5.869/0.023 | 8.138/0.013 | 9.864/0.013 |
| CoT | 0.778/0.181 | 0.407/0.077 | 1.252/0.168 | 0.550/0.134 | 5.239/0.053 | 1.186/0.184 | 2.227/0.082 |
| Self-Refine | 2.317/0.164 | 1.285/0.079 | 4.634/0.245 | 1.802/0.078 | 45.67/0.262 | 2.384/0.023 | 10.43/0.101 |
| *Streaming* | | | | | | | |
| GrowPrompt | 2.819/0.065 | 1.205/0.021 | 3.309/0.069 | 2.103/0.088 | 5.910/0.019 | 7.699/0.012 | 10.56/0.013 |
| MemPrompt | 2.711/0.064 | 1.193/0.021 | 3.178/0.065 | 2.066/0.090 | 5.890/0.018 | 7.932/0.013 | 10.60/0.013 |
| Self-StreamICL | 2.156/0.063 | 0.966/0.021 | 2.688/0.066 | 1.765/0.073 | 5.756/0.019 | 7.833/0.013 | 10.46/0.013 |

Table 11: Cost analysis (using millions of input and output tokens of `gemini-1.0-pro-001`).

| Task | Text-to-SQL | | | Python | Tool Use | Medical | QA |
|------|-------|-------|------|--------|----------|---------|-----|
| Dataset | Spider | CoSQL | BIRD | DS-1000 | ToolBench | DDXPlus | HotpotQA |
| *Non-streaming* | | | | | | | |
| Zero-Shot | 0.871/0.110 | 0.454/0.045 | 1.538/0.114 | 0.591/0.044 | 5.603/0.020 | 1.131/0.011 | 2.222/0.018 |
| Few-Shot | 2.639/0.108 | 1.062/0.043 | 3.668/0.120 | 2.215/0.157 | 6.301/0.023 | 8.553/0.010 | 9.863/0.015 |
| CoT | 0.930/0.322 | 0.477/0.130 | 1.443/0.319 | 0.626/0.158 | 5.626/0.873 | 1.191/0.310 | 2.273/0.083 |
| Self-Refine | 1.944/0.131 | 0.987/0.056 | 2.977/0.129 | 1.869/0.205 | 11.14/0.038 | 2.350/0.019 | 4.587/0.025 |
| *Streaming* | | | | | | | |
| GrowPrompt | 4.307/0.152 | 1.774/0.053 | 4.103/0.109 | 2.146/0.040 | 6.334/0.020 | 8.077/0.010 | 10.82/0.016 |
| MemPrompt | 3.405/0.100 | 1.693/0.049 | 4.012/0.109 | 2.073/0.039 | 6.325/0.020 | 8.354/0.011 | 10.88/0.015 |
| Self-StreamICL | 3.402/0.142 | 1.212/0.034 | 3.299/0.106 | 1.880/0.037 | 6.175/0.019 | 8.250/0.010 | 10.79/0.014 |

Table 12: Cost analysis (using millions of input and output tokens of `claude-3-haiku-20240307`).

| Task | Text-to-SQL | | | Python | Tool Use | Medical | QA |
|------|-------|-------|------|--------|----------|---------|-----|
| Dataset | Spider | CoSQL | BIRD | DS-1000 | ToolBench | DDXPlus | HotpotQA |
| *Non-streaming* | | | | | | | |
| Zero-Shot | 0.911/0.096 | 0.474/0.038 | 1.510/0.094 | 0.669/0.204 | 5.971/0.028 | 1.287/0.011 | 2.419/0.025 |
| Few-Shot | 2.790/0.094 | 1.111/0.036 | 4.047/0.098 | 2.347/0.256 | 6.726/0.027 | 9.160/0.024 | 10.98/0.022 |
| CoT | 0.975/0.391 | 0.500/0.148 | 1.571/0.370 | 0.704/0.341 | 5.996/0.074 | 1.350/0.297 | 2.473/0.166 |
| Self-Refine | 2.088/0.194 | 1.109/0.089 | 3.753/0.195 | 1.624/0.225 | 20.75/0.252 | 2.948/0.298 | 6.427/0.076 |
| *Streaming* | | | | | | | |
| GrowPrompt | 3.562/0.099 | 1.551/0.037 | 4.103/0.098 | 2.336/0.222 | 6.750/0.025 | 8.650/0.022 | 11.77/0.024 |
| MemPrompt | 3.495/0.103 | 1.527/0.038 | 4.067/0.101 | 2.275/0.214 | 6.763/0.025 | 8.924/0.024 | 11.82/0.024 |
| Self-StreamICL | 2.844/0.100 | 1.219/0.036 | 3.409/0.098 | 2.048/0.216 | 6.577/0.026 | 8.666/0.026 | 11.71/0.021 |

Table 13: Cost analysis (millions of input and output tokens) on `gemini-1.5-flash` and `gpt-4o`.

| Task | Text-to-SQL | | | Python | Tool Use | Medical | QA |
|---|---|---|---|---|---|---|---|
| Dataset | Spider | CoSQL | BIRD | DS-1000 | ToolBench | DDXPlus | HotpotQA |
| *gemini-1.5-flash* | | | | | | | |
| Zero-Shot | 0.851/0.103 | 0.439/0.045 | 1.386/0.110 | 0.690/0.048 | 5.603/0.023 | 1.119/0.019 | 2.222/0.018 |
| Self-StreamICL | 2.622/0.092 | 1.274/0.042 | 3.339/0.109 | 2.108/0.045 | 6.201/0.023 | 8.018/0.016 | 10.83/0.013 |
| *gpt-4o-2024-05-13* | | | | | | | |
| Zero-Shot | 0.717/0.079 | 0.377/0.031 | 1.359/0.082 | 0.586/0.067 | 5.171/0.021 | 1.088/0.011 | 2.145/0.019 |
| Self-StreamICL | 2.233/0.076 | 0.978/0.026 | 2.596/0.074 | 1.846/0.060 | 5.705/0.019 | 7.533/0.011 | 10.42/0.013 |

# F    Supplementary materials

The supplementary materials include details such as preprocessing of datasets, prompts of each baseline method, and code to reproduce the experiments.

## F.1    Code repository

The code for reproducing the experiments can be found in our GitHub repository: `https://github.com/stream-bench/stream-bench`.

## F.2    Details for each dataset

We provide the licenses, preprocessing pipelines, calculation of evaluation metrics, and links to the datasets in StreamBench: `https://huggingface.co/datasets/appier-ai-research/StreamBench`. To construct the streaming sequences, one only needs to download the datasets and follow the instructions in our code repository.

Table 14: Licenses of each dataset on StreamBench.

| Task | Text-to-SQL | | | Python | Tool Use | Medical | QA |
|---|---|---|---|---|---|---|---|
| Dataset | Spider | CoSQL | BIRD | DS-1000 | ToolBench | DDXPlus | HotpotQA |
| License | CC BY-SA | CC BY-SA | CC BY-SA | CC BY-SA | Apache-2.0 | CC-BY | CC BY-SA |

### F.2.1    Spider

**Preprocessing**    We download the Spider [17] dataset from their project website: `https://yale-lily.github.io/spider`, and use the original test set as our test set on StreamBench[2].

**Evaluation metric**    We adopt the commonly used Execution Accuracy (EA) for all three Text-to-SQL datasets (Spider, CoSQL, and BIRD). This metric quantifies the proportion of instances where the execution result of the generated SQL $\hat{y}_t$ is identical to that of the ground truth SQL $y_t$ across all instances from time step $t = 1$ to $T$, where $T$ is the number of instances in the test set:

$$\text{EA} = \frac{\sum_{t=1}^{T} \mathbb{1}\left(r_t, \hat{r}_t\right)}{T}$$

Here, $r_t$ represents the execution results of $y_t$, while $\hat{r}_t$ is the execution results of $\hat{y}_t$, with $\mathbb{1}(\cdot)$ being the indicator function defined by:

$$\mathbb{1}(r_t, \hat{r}_t) = \begin{cases} 1, & r_t = \hat{r}_t \\ 0, & r_t \neq \hat{r}_t \end{cases}$$

---

[2]The current "Spider" on StreamBench refers to Spider 1.0. Please refer to their project website for details on the upcoming version of Spider.

### F.2.2 CoSQL

**Preprocessing**  The CoSQL [18] dataset is sourced from its official website: `https://yale-lily.github.io/cosql`. Due to the unavailability of the official test set, we utilize the original development set as our test set on StreamBench. CoSQL was originally formatted in the multi-turn conversation structure, including a sequence of question-SQL pairs (i.e., $(x, y)$ pairs where $x$ is the user's natural language question and $y$ is the SQL code). To adapt CoSQL into the streaming framework of StreamBench, we extract each $(x, y)$ pair from the conversations to build the test set of size $T = 1,007$.

### F.2.3 BIRD

**Preprocessing**  We download the BIRD [19] dataset from their project website `https://bird-bench.github.io/`. Similar to COSQL, we use the full development set as our test set on StreamBench due to the unavailability of the original test set.

### F.2.4 DS-1000

**Preprocessing**  We use the DS-1000 [20] dataset on Huggingface: `https://huggingface.co/datasets/xlangai/DS-1000`. Since this dataset only contains the test set, we manually construct few-shot examples for the few-shot baseline. The few-shot examples are available in our code repository. Please refer to Section F.1.

**Evaluation metric**  DS-1000 adopts pass@1 as the evaluation metric, which denotes the proportion of instances where the agent's code solution $\hat{y}_t$ pass all test cases of $x_t$ for $t = 1, 2, ..., T$.

### F.2.5 ToolBench

**Preprocessing**  Since the original ToolBench [21] contains large-scale real online APIs suffering from instability, we adopt the 50 high-quality APIs curated by STE [22]. Each API has 15 test instances, so there are 750 instances in the test set.

**Evaluation metric**  We following the same evaluation protocol specified in STE [22] to calculate the accuracy. The agent's output $\hat{y}_t$ is considered correct if and only if both the API name and API arguments are correct. The API name is checked by exact string matching. For APIs that have deterministic values for the API arguments, exact string matching is performed. For APIs that accept natural language inputs, a judge LLM is used to evaluate the correctness of API arguments. The implementation details can be found in our code repository (Section F.1).

### F.2.6 DDXPlus

**Preprocessing**  DDXPlus [23] is a large-scale dataset for medical diagnosis, and it contains more than 100,000 instances in the test set originally. Since it would be too expensive to run all test instances on LLMs, we sample equal number of instances from each medical diagnosis to make the test set of size $T = 1,764$ on StreamBench. The full test set is available from the link provided in Section F.2, where $x$ is the patient profile and $y$ is the pathology (i.e., diagnosis). The original dataset can be found in the repository of DDXPlus: `https://github.com/mila-iqia/ddxplus`.

**Evaluation metric**  We use accuracy as the evaluation metric, which is calculated as $\frac{\sum_{t=1}^{T} \mathbb{1}(\hat{y}_t = y_t)}{T}$.

### F.2.7 HotpotQA

**Preprocessing**  We adopt the distractor setting in HotpotQA [24], where each instance contains both supporting or distracting documents for answering the question. The supporting documents have the information needed to answer the question, while the distracting documents do not. Because the test set is not available, we construct the test set by sampling 1,500 instances randomly from the dev set (distractor) downloaded from the official website: `https://hotpotqa.github.io/`.

**Evaluation metric** Following the HotpotQA paper, we adopt exact match (EM) and F1 as two evaluation metrics. We use EM as the primary evaluation metric on StreamBench. However, we also include the calculation of F1 in our code.

### F.3 Prompt templates

We use similar prompt templates in all tasks to minimize prompt engineering. To demonstrate, the prompt templates of the Text-to-SQL task (Spider, CoSQL, and BIRD) as well as the medical diagnosis task (DDXPlus) are provided below. Note that the Self-Refine [26] prompting pipeline involves using the zero-shot prompt to generate the initial output, and then use the *feedback* prompt and *refinement* prompt alternatingly to arrive at the final answer. Therefore, we provide two prompt templates for the Self-Refine baseline, one for feedback and the other for refinement. The prompt templates for other datasets (DS-1000, ToolBench, and HotpotQA) can be found in our code repository.

#### F.3.1 Text-to-SQL

The prompt templates for Spider, CoSQL, and BIRD are provided below.

**Zero-shot**  In this template, {schema} would be replaced by the database schema, while {question} would be replaced by the user's data requirements.

---

{schema}

– Using valid SQLite, answer the following question for the tables provided above.
– Question: {question}

Now, generate the correct SQL code directly (Do NOT generate other text except the SQL code):

---

Figure 11: The prompt template for the zero-shot baseline for Text-to-SQL datasets.

**Chain-of-thought (CoT)**  It is similar to the zero-shot prompt template, except that the trigger phrase "take a deep breath and work on this problem step-by-step to derive the correct SQL code" is appended to the end.

---

{schema}

– Using valid SQLite, answer the following question for the tables provided above.
– Question: {question}

Now, take a deep breath and work on this problem step-by-step to derive the correct SQL code.
Provide your output in the following format:
Rationale: <your_rationale>
Answer: "'sql\n<your_SQL_code>\n"'

---

Figure 12: The prompt template for the CoT baseline for Text-to-SQL datasets.

**Self-Refine**  The feedback prompt and refinement prompt are provided below.

---

You are performing the text-to-SQL task. Here is the database schema, user's question, and your previously generated SQL code.

– SQL schema: {schema}
– User's question: {question}
– Your SQL code: {model_output}

First, determine whether you need to refine your SQL code in terms of its correctness.
If you consider that your SQL code is correct, output 'NO NEED TO REFINE' in uppercase.
Otherwise, provide a suggestion to correct the SQL code.

---

Figure 13: The feedback prompt template for the Self-Refine baseline for Text-to-SQL datasets.

You are performing the text-to-SQL task. Here is the database schema, user's question, and your previous answer-feedback trajectory.

– SQL schema: {schema}
– User's question: {question}
– Your previous answer-feedback trajectory:
{trajectory}

According to the latest feedback, provide your refined SQL code.
Provide your output in the following format:
"'sql\n<your_SQL_code>\n"'

Figure 14: The refinement prompt template for the Self-Refine baseline for Text-to-SQL datasets.

**Prompt template for past information** For the few-shot and streaming methods (GrowPrompt, MemPrompt, Self-StreamICL, and MAM-StreamICL), we use the following prompt template for integrating information of past instances.

You are performing the text-to-SQL task. Here are some examples:

{past_information}

Now it's your turn.

– SQL schema: {schema}
– Using valid SQLite, answer the following question for the SQL schema provided above.
– Question: {question}

Now, generate the correct SQL code directly (Do NOT generate other text except the SQL code):

Figure 15: The prompt template for integrating past information for Text-to-SQL datasets.

**Note** The actual content of {past_information} would be replaced by $k$ templated past instances, and the template is different for different baselines. We provide the templates as follows:

Question: {question}
{sql_code}

Figure 16: The template for each past instance ($k = 16$ in Text-to-SQL) for the few-shot, Self-StreamICL, and MAM-StreamICL baselines.

Question: {question}
Your SQL code: {sql_code}
User Feedback: {verbalized_feedback}

Figure 17: The template for each past instance ($k = 16$ in Text-to-SQL) for the GrowPrompt and MemPrompt baselines. In this template, {verbalized_feedback} is the verbalized $fb_t$, which is "Your answer is correct" when $fb_t = 1$ or "Your answer is not correct" when $fb_t = 0$.

The {past_information} would be replaced by the information of $k$ templated past instances ($k$ varies with datasets, see Section A for details), each delimited by three newlines.

### F.3.2 Medical diagnosis

The prompt templates for DDXPlus are provided below.

**Zero-shot**    The prompt template for the zero-shot baseline on DDXPlus. In this template, {profile} would be replaced by the actual patient profile, while {option_text} would be replaced by all 49 possible diagnoses for the agent to choose from.

---

Act as a medical doctor and diagnose the patient based on the following patient profile:

{profile}

All possible diagnoses for you to choose from are as follows (one diagnosis per line, in the format of <number>. <diagnosis>):
{option_text}

Now, directly provide the diagnosis for the patient in the following format: <number>. <diagnosis>

---

Figure 18: The prompt template for the zero-shot baseline on DDXPlus.

**Chain-of-thought (CoT)**    It is similar to the zero-shot prompt template, except that the trigger phrase "take a deep breath and work on this problem step-by-step to derive the most likely diagnosis" is appended to the end.

---

Act as a medical doctor and diagnose the patient based on the following patient profile:

{profile}

All possible diagnoses for you to choose from are as follows (one diagnosis per line, in the format of <number>. <diagnosis>):
{option_text}

Now, take a deep breath and work on this problem step-by-step to derive the most likely diagnosis. Provide your output in the following valid JSON format:
{"rationale": "<your_rationale>", "answer": "<number>. <diagnosis>"}

---

Figure 19: The prompt template for the zero-shot chain-of-thought (CoT) baseline on DDXPlus.

**Self-Refine**    The feedback prompt and refinement prompt are provided below.

---

You are acting as medical doctor and tasked to diagnose the patient based on the provided patient profile. Here's the patient diagnosis:

{profile}

All possible diagnoses for you to choose from are as follows (one diagnosis per line, in the format of <number>. <diagnosis>):
{option_text}

Your answer : {model_output}

First, determine whether you need to refine your answer in terms of its correctness.
If you consider that your answer is correct, output 'NO NEED TO REFINE' in uppercase.
Otherwise, provide a suggestion to correct the diagnoses in the format of <number>. <diagnosis>.

---

Figure 20: The feedback prompt template for the Self-Refine baseline on DDXPlus.

> You are acting as medical doctor and tasked to diagnose the patient based on the provided patient profile. Here's the patient diagnosis:
>
> {profile}
> All possible diagnoses for you to choose from are as follows (one diagnosis per line, in the format of <number>. <diagnosis>):
> {option_text}
> – Your previous answer-feedback trajectory:
> {trajectory}
>
> According to the latest feedback, provide your new answer
> Provide your output in the following format: one diagnosis per line, in the format of <number>. <diagnosis>

Figure 21: The refinement prompt template for the Self-Refine baseline on DDXPlus.

**Prompt template for past information**   For the few-shot and streaming methods (GrowPrompt, MemPrompt, Self-StreamICL, and MAM-StreamICL), we use the following prompt template for integrating information of past instances.

> Act as a medical doctor and diagnose the patient based on the provided patient profile.
>
> All possible diagnoses for you to choose from are as follows (one diagnosis per line, in the format of <number>. <diagnosis>):
> {option_text}
>
> Here are some example cases.
>
> {past_information}
>
> Now it's your turn.
>
> {profile}
>
> Now provide the diagnosis for the patient in the following format: <number>. <diagnosis>

Figure 22: The prompt template for multiple baselines on DDXPlus.

**Note**   The actual content of {past_information} is different for different baselines:

> {profile}
> Diagnosis: {diagnosis}

Figure 23: The template for each past instance ($k = 16$ in DDXPlus) for the few-shot, Self-StreamICL, and MAM-StreamICL baselines.

> {profile}
> Your answer: {diagnosis}
> User Feedback: {verbalized_feedback}

Figure 24: The template for each past instance ($k = 16$ in DDXPlus) for the GrowPrompt and MemPrompt baselines. In this template, {verbalized_feedback} is the verbalized $fb_t$, which is "Your answer is correct" when $fb_t = 1$ or "Your answer is not correct" when $fb_t = 0$.

The {past_information} would be replaced by the information of $k$ templated past instances ($k$ varies with datasets, see Section A for details), each delimited by three newlines.

## F.4 Other details

### F.4.1 Algorithm of ablation studies with MAM-StreamICL

In Table 9, we conduct ablation studies on MAM-StreamICL to show the importance of multi-agent memory. The algorithm of "MAM-StreamICL (ablation)" in this table is provided as follows:

---

**Algorithm 3** Algorithm for MAM-StreamICL (ablation)

---

1: Initialize $K$ agents $f_0(\cdot|\theta_0), f_1(\cdot|\theta_1), ..., f_{K-1}(\cdot|\theta_{K-1})$; ▷ K = 1 in the Self-StreamICL baseline
2: Initialize prompt $p(\cdot)$, retriever $r(\cdot)$, and external memory $\mathcal{M}_0$, all shared between agents;
3: Select an agent $f_s(\cdot|\theta_s)$ as the source of single-agent memory; ▷ For example, we can choose `gemini-1.0-pro-001`
4: **for** $t = 1, 2, \ldots, T$ **do**
5:  Receive instance $x_t$ from the data stream;
6:  Select the next agent by $k = t \bmod K$;
7:  The $k$-th agent predicts $\hat{y}_t = f_k(p(x_t, r(\mathcal{M}_{t-1}))|\theta_k)$;   ▷ $\hat{y}_t$ is used to for evaluation
8:  The chosen single agent predicts $\hat{y}_{t_s} = f_s(p(x_t, r(\mathcal{M}_{t-1}))|\theta_s)$; ▷ Counterfactual ablation
9:  Receive feedback signal $fb_{t_s} = g(x_t, \hat{y}_{t_s})$;      ▷ $\hat{y}_{t_s}$ is used for receiving feedback
10:  **if** $fb_{t_s} = 1$ **then**        ▷ which means the self-output $\hat{y}_t$ is correct
11:    $\mathcal{M}_t \leftarrow \mathcal{M}_{t-1} \cup \{(x_t, \hat{y}_{t_s})\}$;
12:  **else**
13:    $\mathcal{M}_t \leftarrow \mathcal{M}_{t-1}$;
14:  **end if**
15: **end for**

---

The parts different from the original MAM-StreamICL algorithm is highlighted in red. This ablated algorithm can be seen as a counterfactual experiment, where we use multiple agents for *inference* but only one chosen agent for the *memory* mechanism. The results in Table 9 show that both multi-agent memory and multi-agent inference are beneficial for performance boost.

