# OpenReview forum: "StreamBench: Towards Benchmarking Continuous Improvement of Language Agents"
_NeurIPS.cc/2024/Datasets_and_Benchmarks_Track — NeurIPS 2024 Track Datasets and Benchmarks Poster_

### Official Review · Reviewer_GLtH · 2024-07-23

**Rating:** 7
**Confidence:** 3
**Correctness:** As described in the review text.
**Clarity:** Yes, the paper is well written and ea…

**Review:**

Overall, the paper is easy to follow and is a positive first step towards evaluating LLM agent’s abilities to improve in-context given feedback.

Pros:

The paper is well written and easy to follow.
The authors consider a variety of datasets and non-streaming methods for their benchmark and clearly show how the streaming method can improve performance.
The framework is general and can be applied to improve various aspects of LLM agents.
Showing how the multi-agent round robin stream based algorithm (MAM-StreamICL) can be used to combine the strengths of 3 LLM agents while maintaining the average cost of only 1 agent, as shown in FIgure 3.
The authors have illustrated that streaming is an effective means of improving the performance of both smaller and larger LLM agents.

Cons:

The code repo is not yet finalised.
The sequences for ordering the datasets are not easy to find.

**Strengths:**

As far I am aware, StreamBench is the first LLM agent benchmark of its kind which could open a new avenue for investigating how well LLM agents could improve over time. This could lead to the further development of more real-world accurate benchmarks for LLMs and LLM agents.

**Additional Feedback:**

How were the 1500 Question Answering dataset questions sampled? Is this subset of questions available for future research?

Did the authors consider more non-streaming methods for the benchmarks on gpt-4o and gemini-1.5-flash? If so, why were these results not reported?

**Documentation:**

The datasets are publicly available and downloadable. All dataset licenses, preprocessing pipelines and evaluation metrics calculation details are given. Datasets are also hosted on Huggingface and the code is publicly available.

**Limitations:**

Yes, the authors have mentioned the limitations of their benchmark.

**Opportunities For Improvement:**

The results have no error quantification in Figure 1 and Table 2. It would be useful to have given the errors from different sequence orderings here, and in Figure 4.

The suggested algorithms for LLM agent improvement by the authors Self-StreamICL and MAM-StreamICL both require a binary feedback signal. It would be interesting to see how a different feedback signal can be handled.

In Table 3 when larger models are considered more non-streaming methods can be used.

The repo can be improved and an interactive notebook can be made for evaluating, even a small LLM agent, on streambench.

It would be interesting if the authors could develop some dataset where knowledge of previous outputs not only helps an LLM agent improve but is required. In all the benchmarks mentioned this is not the case.

**Relation To Prior Work:**

As far as I am aware a streaming benchmark for LLMs does not exist. The authors do mention that there are approaches for improving certain aspects of LLM agents.
The authors also mention existing streaming methods and discuss their weaknesses.

**Summary And Contributions:**

The authors introduce StreamBench which is a benchmark for measuring how LLM agents can improve their capabilities in context over multiple consecutive timesteps on a particular task. This is in contrast to the standard way of evaluating LLM agents which is to only consider the innate capabilities of a given LLM agent one timesteps at a time with static datasets. The authors augment 5 datasets to be stream-based which form the base for StreamBench. In order to make datasets stream-based, authors generate and make available dataset sequences to go along with these datasets for agent evaluation. Additionally the authors introduce some techniques for improving stream-based LLM agents; one of which is a cost effective multi-agent round-robin style prompting technique which can leverage the strengths of multiple LLM agents to self improve on a given task. The authors convert five known datasets to be streaming datasets and make the streaming sequence orderings of five predetermined random seeds publicly available for future research.

---

> ### Author Rebuttal · Authors · 2024-08-17
>
> Thank you for your warm feedback for recognizing StreamBench's potential impacts on advancing stream-based LLM agents. We also appreciate your comments on how to improve our work. To follow your suggestions:
> > The code repo is not yet finalised. The sequences for ordering the datasets are not easy to find.
> * We now finalize the code repo, and add an interactive notebook `obtain_streaming_sequences.ipynb` about how to obtain the sequences of ordering the datasets.
>
> > The results have no error quantification in Figure 1 and Table 2. It would be useful to have given the errors from different sequence orderings here, and in Figure 4.
> * To ensure we correctly understand your suggestion, we provide error quantification for `gpt-3.5-turbo-0125` as an example below, with standard errors shown in parentheses:
> |Method / Dataset|BIRD|DS-1000|ToolBench|DDXPlus|HotpotQA|
> |------------------|----------------|-----------------|-----------------|----------------|----------------|
> |GrowPrompt| 30.08% (0.33%) | 25.36% (0.45%)  | 67.92% (0.41%)  | 44.47% (0.50%) | 55.52% (0.19%) |
> |MemPrompt| 32.85% (0.68%) | 28.50% (1.43%)  | 67.33% (1.33%)  | 51.03% (0.70%) | 56.26% (0.14%) |
> |StreamICL| 34.98% (0.68%) | 39.54% (0.62%)  | 74.11% (0.86%)  | 64.89% (1.06%) | 56.12% (0.42%) |
> * If this presentation aligns with your expectations, we will include the standard errors from different sequence orderings in Table 2 and Figure 4, and add shading to the performance curves in Figure 1 to better illustrate the error margins.
>
> > The suggested algorithms for LLM agent improvement by the authors Self-StreamICL and MAM-StreamICL both require a binary feedback signal. It would be interesting to see how a different feedback signal can be handled.
> * We agree that exploring how different types of feedback signals can be handled is a promising direction. We briefly touched on this in our paper (lines 92-96 and in the limitations section) and would be glad to pursue this research direction with the community.
>
> > In Table 3 when larger models are considered more non-streaming methods can be used.
>
> > Did the authors consider more non-streaming methods for the benchmarks on gpt-4o and gemini-1.5-flash? If so, why were these results not reported?
> * We did consider adding more non-streaming methods for `gpt-4o` and `gemini-1.5-flash` in Table 3. However, due to budget constraints at the time of writing the paper, we were unable to conduct these experiments.
> * Now that we have secured additional budgets, we are pleased to provide the results below, with `gpt-4o-2024-05-13` replaced by `gpt-4o-2024-08-06` for lower cost. The findings remain consistent with our original paper.
>
> |`gemini-1.5-flash-001`| Spider | CoSQL | BIRD  | DS-1000 | ToolBench | DDXPlus | HotpotQA |
> |----------------------|--------|-------|-------|---------|-----------|---------|----------|
> | Zero-shot            | 69.63% | 48.26%| 33.83%| 50.20%  | 69.47%    | 58.90%  | 60.60%   |
> | Few-shot             | 71.40% | 49.35%| 37.03%| 50.60%  | 72.13%    | 73.58%  | 64.87%   |
> | CoT                  | 72.52% | 52.73%| 35.14%| 44.80%  | 68.00%    | 64.06%  | 63.13%   |
> | Self-StreamICL       |**77.83%**|**56.21%**|**41.20%**|**52.20%**|**75.07%**|**86.34%**|**65.20%**|
>
> |`gpt-4o-2024-08-06`| Spider | CoSQL | BIRD  | DS-1000 | ToolBench | DDXPlus | HotpotQA |
> |--------------------|--------|-------|-------|---------|-----------|---------|----------|
> | Zero-shot          | 73.54% | 53.33%| 34.42%| 54.90%  | 72.40%    | 70.64%  | 65.53%   |
> | Few-shot           | 76.85% | 57.60%| 36.25%| 52.30%  | 71.47%    | 83.45%  | 66.87%   |
> | CoT                | 72.52% | 54.82%| 31.16%| 41.90%  | 66.80%    | 73.02%  | 62.80%   |
> | Self-StreamICL     |**80.58%**|**59.19%**|**42.63%**|**59.40%**|**76.27%**|**92.01%**|**67.00%**|
> * Please note that we did not include the Self-Refine baseline for `gpt-4o` and `gemini-1.5-flash`, as it is too expensive (as mentioned in **Table 8**) and did not yield performance improvements over other non-streaming baselines.
>
> > The repo can be improved and an interactive notebook can be made for evaluating, even a small LLM agent, on streambench.
> * Thank you for the suggestion! We now include an interactive notebook `playground.ipynb` as a step-by-step walkthrough about how to run StreamBench.
> * For evaluating other LLM agents (including the smaller ones), we now support HuggingFace models (implemented in `./stream_bench/llms/hf_model.py`). As an example, one can run `./scripts/run_ddxplus.sh` to evaluate `google/gemma-2-2b-it` on StreamBench)
>
> > It would be interesting if the authors could develop some dataset where knowledge of previous outputs not only helps an LLM agent improve but is required. In all the benchmarks mentioned this is not the case.
> * Thank you for the suggestion! A future direction we are considering involves curriculum learning (Bengio et al., 2009), where instances are presented in a meaningful order that progressively introduces more complex concepts, such as moving from simple arithmetic (e.g., GSM8K) to competition-level math (e.g., MATH by Hendrycks et al.).
>
> > How were the 1500 Question Answering dataset questions sampled? Is this subset of questions available for future research?
> 1. These 1500 questions were randomly sampled from the distractor dev set from the original HotpotQA dataset.
> 2. Yes, these sampled questions are hosted on our Huggingface dataset page and available for future research.
> 3. We acknowledge that both points 1 and 2 are currently mentioned in the supplementary materials, which may be hard to notice. We will reorganize our paper to make this information more easily accessible to readers.

---

> ### Author Response · Authors · 2024-08-24
> **Follow-Up on Review Feedback**
>
> As the deadline of author-reviewer discussion period approaches, we wanted to follow up on our responses to your feedback. Could you please let us know if our responses have addressed your questions? Thank you very much.

---

> > ### Comment · Reviewer_GLtH · 2024-08-26
> > **Response to author feedback**
> >
> > I would like to thank the authors for the answers, the additional information and the added experiments. I would thus like to upgrade my rating accordingly.

---

> > > ### Author Response · Authors · 2024-08-26
> > > **Response to Reviewer GLtH's Feedback**
> > >
> > > Thank you for your thoughtful consideration of our responses. We greatly appreciate your positive update.

---

### Official Review · Reviewer_XjJz · 2024-07-27

**Rating:** 6
**Confidence:** 2
**Correctness:** Yes
**Clarity:** Yes

**Review:**

### Strengths:
* This work makes pioneering contributions by addressing the lack of benchmarks that evaluate an LLM agent's ability for continuous improvement, a gap that remains despite the extensive benchmarks for measuring LLM agents' innate capabilities.
* The writing is clear and easy to follow.

### Weaknesses and opportunities for improvement
* My biggest concern is about the streaming sequence paragraph in Section 3.1. What's the motivation behind randomly (but with a fixed seed) assigning a time step to each instance? This sounds like a pretty arbitrary design to me. Does it make sense? How closely dose it resemble the real scenarios?
* How do the authors implement the feedback signals for datasets considered in StreamBench?

**Strengths:**

Please see the review above.

**Additional Feedback:**

None

**Documentation:**

Yes

**Limitations:**

Yes, they discuss the limitations in Section 8.

**Opportunities For Improvement:**

Please see the review above.

**Relation To Prior Work:**

Yes

**Summary And Contributions:**

This paper presents StreamBench, a benchmark designed to evaluate LLM agents' ability to continuously enhance their performance over time. Specifically, StreamBench simulates an online learning environment where the LLM agent can improve its response based on an input-feedback sequence. The authors also propose several simple baselines for enhancing LLM agents in such streaming scenarios and provide insights into effective streaming strategies for LLMs.

---

> ### Author Rebuttal · Authors · 2024-08-17
>
> Thank you for recognizing the pioneering contributions of our work. We greatly appreciate your constructive comments, and are happy to address them to further improve our work:
>
> > My biggest concern is about the streaming sequence paragraph in Section 3.1. What's the motivation behind randomly (but with a fixed seed) assigning a time step to each instance? This sounds like a pretty arbitrary design to me. Does it make sense? How closely does it resemble the real scenarios?
> * Thank you for raising this point. The motivation behind randomly assigning time steps is to simulate scenarios where each instance in the streaming sequence is drawn from the same distribution. This design is intended to reflect scenarios without distributional shifts, and our current experiment results are relevant to such cases.
> * We use a fixed random seed to ensure reproducibility for future researchers. To address the variance issue with a single seed, we also reran the streaming experiments with five random seeds (Figure 4 in our paper), where the findings remain consistent.
> * However, we agree that it is important to investigate streaming sequences with distributional shifts. To this end, we conduct experiments on the text-to-SQL dataset **BIRD** (Jinyang et al., 2024) as an example:
>
> | Method/LLM | gpt-3.5-turbo-0125 | gemini-1.0-pro-001 | claude-3-haiku | gemini-1.5-flash-001 | gpt-4o-2024-08-06 |
> |-------------------------------------|--------------------|--------------------|---------------|---------------------|-------------------|
> | Zero-Shot | 29.75%             | 28.16%             | 30.90%        | 33.83%              | 34.42%            |
> | Few-Shot | 28.94%             | 31.49%             | 30.77%        | 37.03%              | 36.25%            |
> | CoT | 29.21%             | 21.71%             | 30.77%        | 35.14%              | 31.16%            |
> | Self-StreamICL **(no distributional shifts)** | 35.07%             | 33.25%             | 37.61%        | 41.20%              | 42.63%            |
> | Self-StreamICL **(with distributional shifts)** | 36.31%             | 32.60%             | 36.57%        | 40.48%              | 43.09%            |
> * In the **BIRD** test set on StreamBench, there are 11 different databases, each containing around one hundred instances. To simulate distributional shifts, instances from the same database are arranged consecutively (DB1 -> DB2 -> … -> DB11), resulting in 10 distribution shifts during the simulation.
> * The key finding that Self-StreamICL outperforms non-streaming baselines remains consistent with our paper. However, we fully agree that incorporating real-world streaming data will be an important future direction.
> * We would be happy to provide experiments on other datasets if you’re interested.
>
> > How do the authors implement the feedback signals for datasets considered in StreamBench?
> * **High-level overview:** At each time step $t$, the feedback signal is “1” if the agent’s output $\hat{y}_t$ is “correct”, and “0” if $\hat{y}_t$ is “incorrect”. The actual implementation depends on how “correctness” is defined by the original datasets, which we elaborate as follows:
>   - **Text-to-SQL datasets (Spider, CoSQL, and BIRD):** In these three datasets, $\hat{y}_t$ is the agent’s generated SQL code, and $y_t$ is the ground truth SQL code. The “correctness” is defined by SQL execution accuracy. That is, whether the execution results of $\hat{y}_t$ and $y_t$ are the same.
>   - **Python programming (DS-1000):** Given a question in DS-1000, the “correctness” is defined by whether the generated python code $\hat{y}_t$ passes all the test cases for that question.
>   - **Tool use (ToolBench):** The “correctness” is defined by whether both the “API name” and “API arguments” are correct in $\hat{y}_t$. Otherwise, it is considered incorrect.
>   - **Medical diagnosis (DDXPlus):** Whether the predicted diagnosis $\hat{y}_t$ is the same as the ground truth diagnosis $y_t$.
>   - **Question answering (HotpotQA):** The “correctness” is defined by exact match. That is, the generated answer text must be the same as the ground truth answer text. Note that this implementation might be relaxed if the answer doesn’t have to be strictly the same for a given application.
>
> * After receiving feedback signals of “1” or “0,” it is up to the design of an agent / baseline method to decide how to utilize them. For example, in GrowPrompt and MemPrompt, we verbalized “1” as “Your answer is correct” and “0” as “Your answer is incorrect” to inform LLM agents about the accuracy of their outputs.

---

> ### Author Response · Authors · 2024-08-24
> **Follow-Up on Review Feedback**
>
> As the author-reviewer discussion phase is about to come to an end, could you please let us know if our responses have addressed your concerns? Thank you very much for your time.

---

> ### Author Response · Authors · 2024-08-31
> **Kind Reminder of Author-Reviewer Discussion Deadline**
>
> We hope you are doing well. We would like to kindly remind you that the deadline for the reviewer-author discussion is approaching, with only 1 day and a few hours remaining. We value your feedback and want to know if our responses have addressed your concerns. Thank you for your time.

---

### Official Review · Reviewer_x115 · 2024-07-31
**Review of 1158**

**Rating:** 7
**Confidence:** 3

**Review:**

The paper seems to have an original idea and to the best of my knowledge I am not aware of any similar endeavours. This work is quite significant in order to gauge in the contextual adaptation of LLMs. Considering, there is similar work/analysis that points to the efficacy of in-context learning vs finetuning, this work provides a basis to further analyse the same. The paper is well written in a simple language although there are some changes I'd like to see. Overall, this work is relevant to the community.

**Strengths:**

1. The paper is straightforward about its goal and its motivations for the same
2. The authors compare multiple types of baselines
3. The benchmark itself covers domains that LLMs are currently used heavily for, namely, programming, tool use and QA.
4. Further adds to the evidence that LLMs are capable of in-context learning and we should be exploring alternative methods of eliciting the same capability more efficiently and consistently

**Additional Feedback:**

None

**Clarity:**

Overall, the paper is quite clear. Although, I'd appreciate more information on how the memory structure functions with multiple different LLMs working in tandem - this point particularly was unclear.

**Correctness:**

The paper appears to be correct for the most part. I'd like the authors to comment on the correctness and quality of the medical datasets they've chosen for this work.
I am also interested in knowing whether the baselines were tuned to the best of the authors' abilities as well as the exclusion of certain finetuning and guidance methods.

**Documentation:**

There was no URL provided in the paper. However, I did find their project through a quick Google search. The repository doesn't appear to be complete yet for reproducing some results. There is a lack of documentation with regards to how one could utilise this benchmark with their own models.

**Ethics:**

No major concerns.

**Limitations:**

The authors correctly point out the lack of multi-modality and task coverage. As it stands, the tasks don't cover all modalities and a broader variety of tasks that LLMs are being used for. I'd like to add to this by pointing out that the benchmark doesn't seem to support multi-lingual tasks which could be an interesting evolution.

**Opportunities For Improvement:**

1. I'd have liked to see an example of the prompt template in the main paper itself
2. It is not clear how the memory is implemented. It appears that the text embeddings are being shared between multiple types of LLMs - how are they able to share the same embeddings? This is not particularly clear.
3. An analysis of the embeddings in memory should lend more insight into the performance as well.
4. I'd like to see a comment on how well tuned the baselines were (if applicable).
5. It is not clear why the authors exclude LORA based finetuning methods or methods similar to DSPy or TextGrad.

**Relation To Prior Work:**

Yes, this is discussed.

**Summary And Contributions:**

- Authors identify a gap in the existing literature - there are no benchmarks to evaluate LLM agents in an online continual learning environment
- Therefore, they propose a new benchmark for online evaluation of LLM agents where agents attempt to solve downstream tasks
- Authors also provide a set of baselines and an analysis of their online continual learning method that uses multiple agents, shared memory and in context learning/adaptation.

---

> ### Author Rebuttal · Authors · 2024-08-17
>
> Thank you for the recognition of our contributions and the insightful comments. We would be very glad to improve our work based on your suggestions as follows:
>
> > I'd have liked to see an example of the prompt template in the main paper itself
> * Thank you for the suggestion. We plan to utilize the additional page in the camera-ready version to provide this content if our paper is accepted.
>
> > It is not clear how the memory is implemented. It appears that the text embeddings are being shared between multiple types of LLMs - how are they able to share the same embeddings?
>
> > I'd appreciate more information on how the memory structure functions with multiple different LLMs working in tandem.
> * Thank you for pointing this out. To clarify, we use the same encoder `BAAI/bge-base-en-v1.5` to generate text embeddings for storing past instances in the memory, which is shared between multiple LLMs. During retrieval, we encode input $x_t$ with the same encoder and retrieve $k$ closest (lowest L2 distance) instances in the memory. We will add these details to our paper.
>
> > An analysis of the embeddings in memory should lend more insight into the performance as well.
> * We agree that analyzing how text embeddings correlate with performance is important. To explore this, we have included results on DDXPlus using different text encoders beyond `bge-base-en-v1.5`:
>
> | Method (`encoder`) / LLM | gpt-3.5-turbo-0125 | claude-3-haiku | gemini-1.5-flash-001 |
> |--------------------------------------------|--------------------|---------------|---------------------|
> | Self-StreamICL w/ `all-MiniLM-L6-v2` (22.7M) | 63.61%| **78.91%**|83.50%|
> | Self-StreamICL w/ `bge-small-en-v1.5` (33.4M)| 63.55%| 75.51%|83.90%|
> | Self-StreamICL w/ `bge-base-en-v1.5` (109M)|**66.16%**| 76.02%|**86.34%**|
>
> * We observe that within the same text encoder family (`bge`), the larger model (109M parameters) generally delivers better performance. However, smaller models (22.7M parameters) can also achieve strong results, indicating that each LLM may benefit from a specific encoder. If you would like to see experiments on other datasets or encoders, we would be happy to provide them.
>
> > I'd like to see a comment on how well tuned the baselines were (if applicable).
>
> > I am also interested in knowing whether the baselines were tuned to the best of the authors' abilities.
>
> * When implementing the baselines, we manually inspect the outputs of each baseline with each LLM. Based on the inspected outputs, we tune the baselines to the points where there were no naive mistakes, such as formatting or parsing errors. Therefore, the difference in performance between the methods arise from the actual difference of the generated content’s quality.
> * Since LLMs are very sensitive to prompts, we ran experiments with different task instructions to see whether the experiment findings remain consistent. We present experiments on **BIRD** with `gemini-1.5-flash-001` to demonstrate:
> | Method  | Prompt 1 | Prompt 2 | Prompt 3 | Mean | Std |
> |---------------------------|--------------------|--------------------|--------------------|--------|-------|
> | Zero-shot | 33.83% | 34.03%| 35.14% | 34.33% | 0.70% |
> | Few-shot | 37.03% | 37.16% | 37.48% | 37.22% | 0.23% |
> | CoT | 35.14% | 34.94% | 33.31% | 34.46% | 1.00% |
> | Self-StreamICL | **41.20%**| **39.96%** | **41.07%**| **40.74%** | 0.68% |
>
> * While there are variations in performance depending on the task instructions used, the relative ranking of methods remains consistent.
>
> > It is not clear why the authors exclude LORA based finetuning methods or methods similar to DSPy or TextGrad.
> * We initially excluded LORA-based fine-tuning to focus on in-context adaptation of LLMs. However, we agree that including baselines like LORA, DSPy, and TextGrad would offer valuable insights. Here are the experiment results for LORA-based fine-tuning on `llama-3-8b-instruct (4 bit)`:
>
> | Method | Spider | CoSQL | BIRD  | DS-1000 | ToolBench | DDXPlus | HotpotQA |
> |--------------------------|--------|-------|-------|---------|-----------|---------|----------|
> |Zero Shot|56.69| 45.38 |18.97|17.3| 36.67| 32.88|49.67|
> |Zero Shot + Lora rank 64|61.48|47.17 |20.67|18.7|53.46|48.58|54.67|
>
> * Following the setting on StreamBench, we update the model at each time step when the feedback signal = 1 (correct). A promising direction is to develop more sophisticated fine-tuning methods enhance performance on StreamBench.
>
> > I'd like the authors to comment on the correctness and quality of the medical datasets they've chosen for this work.
> * The medical dataset we used, DDXPlus (Tchango et al., 2022), is synthesized from a proprietary medical knowledge base and guided by assumptions reviewed by medical doctors.
> * We chose DDXPlus because it offers the most extensive coverage of symptoms and diagnoses among all automatic diagnoses datasets that we know of, making it the closest to real clinical scenarios, though still not fully representative.
> * Regarding correctness and quality, most patient profiles in DDXPlus reasonably map symptoms to diagnoses. However, we do identify a small number of inconsistencies, so curating a higher-quality subset of DDXPlus would be beneficial.
>
> > There was no URL provided in the paper. However, I did find their project through a quick Google search. The repository doesn't appear to be complete yet for reproducing some results. There is a lack of documentation with regards to how one could utilise this benchmark with their own models.
> * The URLs are currently provided in the supplementary materials. We will include them in the main text for readers to find more easily.
> * The code repository is now complete for reproducing all results.
> * We now support HuggingFace models (implemented in `./stream_bench/llms/hf_model.py`), so that future researchers are able to run their own models. We also include documentation on how to run customized LLMs in the README (see “Steps to Run Your Own LLMs”).

---

> ### Author Response · Authors · 2024-08-24
> **Follow-Up on Review Feedback**
>
> As the author-reviewer discussion phase is approaching to an end, we would like to follow up on our responses to your feedback. Could you please let us know if our responses have addressed your comments? Thank you very much for your time.

---

> > ### Comment · Reviewer_x115 · 2024-08-27
> > **Reply**
> >
> > I'd like to thank the authors for their responses and for running additional experiments. I have updated my rating.

---

> > > ### Author Response · Authors · 2024-08-28
> > > **Response to Reviewer x115's Feedback**
> > >
> > > Thank you for thoughtfully considering our responses. We appreciate your positive update.

---

### Author Rebuttal · Authors · 2024-08-17

# Global Rebuttal
We sincerely thank all the reviewers for their precious time and thoughtful feedback on our work. We greatly appreciate your recognition of our pioneering contributions in measuring the continuous adaptation and improvement of LLM agents. Below, we summarize the key strengths, comments and suggestions raised by the reviewers, along with our responses.

## Strengths
* **Pioneering contributions:** StreamBench is the first benchmark to evaluate LLM agents’ ability for continuous improvement, addressing the gap in the existing works.
* **Coverage:** This work contains a variety of baselines and datasets, including the ones LLMs are currently used heavily for.
* **Clarity:** Reviewers found the paper well-written and easy to follow.

## Comments, Suggestions, and Our Responses
* **Incomplete code repository:** our codebase is not complete originally
  - The code repository is now complete
* **Documentation issues:** the prompt templates and URLs to the codecase and datasets were included in the supplementary materials, which could be difficult to be found
  - We will move the prompts and URLs to the main paper for better visibility
* **Further experiments:** thanks to the reviewers for suggesting the following experiments or further analysis for improving our work
  - Relationship between LLM agents’ performance and text embeddings in agents’ memory can provide more insights
    * We conducted experiments with three different text encoders and found that (1) within a given encoder family, larger models generally yield better results, and (2) each LLM benefits from a specific text encoder, even if it’s smaller in size.
  - Whether the baselines were well-tuned
    * We carefully inspected model outputs during baseline implementation, ensuring that naive mistakes, such as formatting or parsing errors, were minimized.
    * To address prompt sensitivity, we conducted experiments with different task instructions across baselines and confirmed that our findings remained consistent.
  - Fine-tuning based methods
    * In our paper, we excluded fine-tuning-based methods to focus on the in-context adaptation of LLMs. However, we agree that fine-tuning approaches should be explored and compared to in-context methods.
    * To see whether fine-tuning based methods can increase performance in the streaming setting, we now provide experiments with LORA fine-tuning on `llama-3-8b-instruct (4 bit)`
  - Motivation behind randomly assigning time steps
    * The random assignment is intended to simulate scenarios where each instance is drawn from the same distribution.
    * We have now included experiments with streaming under distributional shifts and found that our key findings remain consistent (Self-StreamICL > non-streaming baselines).
  - Adding more non-streaming baselines to the latest models `gpt-4o` and `gemini-1.5-flash`
    * We did not run these baselines due to budget constraints at the time
    * We have now conducted the experiments, and the results show that our findings remain consistent.
* **Future directions**
  - Multi-modality and task coverage
    * We also touched on this in the limitations section
  - Multi-lingual tasks
    * Thanks to reviewer `x115` for adding this valid point, which we would certainly like to include in the future
  - Alignment to real scenarios
    * Thanks to reviewer `XjJz` for bringing this up. We agree that incorporating real streaming data is important.
  - Developing datasets where knowledge of past outputs not only helps LLM agents but are required in the future
    * Thanks to reviewer `GLtH` for this suggestion. Exploring curriculum learning (Bengio et al., 2009) as a future direction would indeed be very interesting.

## Final Remarks
We deeply appreciate these valuable suggestions, which are crucial for refining our work and advancing this new research direction.

---

### Author Response · Authors · 2024-08-27
**Author Messages to AC**

Dear ACs,

We would like to bring to your attention that, as of now, reviewers `x115` and `XjJz` have not yet engaged in the author-reviewer discussions or responded to our rebuttal. We have provided detailed responses to their comments, and would be glad to receive further feedback on the results we've submitted.

We kindly request your assistance in following up with the reviewers to ensure that their feedback is incorporated into the final evaluation of our submission.

Thank you for your support.

---

### Author Response · Authors · 2024-09-01
**Summary of Author-Reviewer Discussion**

We would like to express our gratitude to the Area Chair (AC) for assisting in following up with the reviewers during the discussion period. We also thank reviewers `GLtH` and `x115` for engaging in the discussion and providing positive feedback on our rebuttal.

We deeply appreciate all reviewers’ comments, as they help refine our work and further advance this line of research. Although reviewer `XjJz` has not yet responded to our rebuttal, we have endeavored to address the raised concerns through explanations and additional experiments.

---

### Decision · Program_Chairs · 2024-09-26

**Decision:**

Accept (Poster)

**Comment:**

In this paper, the authors propose a benchmark for evaluating LLM agents in an online continual learning setting post deployment. The work has significant implications towards democratic advancement of large language models given the cost and sustainability of periodically updating model parameters.

I concur with the reviewers tat the paper is well written, clear, and easy to follow. The reviewers raised valid concerns, especially about the computation and analysis of memory embeddings, comparisons against additional non-streaming methods, and evaluating streaming sequences with distributional shifts. The authors have provided a comprehensive rebuttal addressing the concerns with appropriate empirical evidence.

Given the comprehensive and thorough updated analysis, the significance of this work toward democratic development of LLMs, and the overall quality of the manuscript, I recommend accepting this paper as it would be a valuable addition to the program.

I encourage the authors to use the extra page to incorporate the results they have shared in the rebuttal into the main paper.